# The Current State of Adult Glial Tumor Patients’ Care in Kazakhstan: Challenges in Diagnosis and Patterns in Survival Outcomes

**DOI:** 10.3390/biomedicines11030886

**Published:** 2023-03-13

**Authors:** Aisha Babi, Karashash Menlibayeva, Torekhan Bex, Shynar Kuandykova, Serik Akshulakov

**Affiliations:** 1Hospital Management Department, National Centre for Neurosurgery, 34/1 Turan Avenue, Astana 010000, Kazakhstan; 2Department of Vascular and Functional Neurosurgery, National Centre for Neurosurgery, 34/1 Turan Avenue, Astana 010000, Kazakhstan

**Keywords:** glial tumor, survival rate, glioblastoma, healthcare disparities, histology, tumor classification

## Abstract

Background: The study aimed to analyze the 5-year survival of adult patients with glial tumors and to define characteristics that are associated with the disease outcomes in Kazakhstan. Methods: Medical records of patients that were surgically treated at the National Center for Neurosurgery during the 5-year period from 2016 to 2020 were collected retrospectively. Patients with a histologically confirmed diagnosis of diffuse astrocytic or oligodendroglial tumor type were included and their survival was assessed with life tables, Kaplan–Meier plot, and Cox regression using STATA 16 statistical software. Results: Almost half of the patients had glioblastoma. The 5-year survival rate of the whole sample was 45.93%. Among Grade 4 patients, 15.6% survived the 5-year mark. Differences in survival between grades 1–3 were not significant. Grade 1 patients demonstrated worse survival rates compared to Grade 2 patients (69% vs. 74%). Worse survival rates were observed among patients of Russian ethnicity and in rural residents. Conclusions: The study described the unusual patterns in survival rates of glial tumor patients in Kazakhstan, pointing to the need for reassessment of diagnostic accuracy and resulting treatment of glial patients in Kazakhstan, and the need to introduce molecular and genetic parameters in tumor type classification. Moreover, the observed difference in survival of different ethnic groups and residents of rural and urban areas should be further investigated and addressed by healthcare professionals.

## 1. Introduction

Approximately 300 thousand people globally were diagnosed with primary brain and spinal cord tumors in 2020 [1]. Of all brain and central nervous system tumors, about 30% account for malignant tumors, of which the majority (48.6%) are glioblastomas, 11.8% are diffuse/anaplastic astrocytomas, and 17.9% account for other gliomas [2]. The incidence rate in the USA for malignant brain and other central nervous system tumors is 7.08 per 100 thousand population, ranging from 8.30 in men to 6.01 in women, with the majority of incidence occurring in 65 years and older demographic within both genders [3]. Information on the incidence of malignant brain tumors in Kazakhstan is limited, however, it has been reported that the incidence of malignant(central nervous system) CNS tumors increased in the period from 2004 to 2011 [4].

Malignant gliomas are the most frequent primary brain tumors of the central nervous system, characterized by their aggressive nature and the limited number of curative treatment options that are currently available. Standard treatment methods include surgical removal followed by radiotherapy, chemotherapy, second-line anticancer treatment, and palliative care. Despite advances in diagnostic methods and improvements in treatment options, the mortality rate for patients with malignant gliomas remains among the highest.

The survival time after diagnosis with malignant brain tumors varies depending on the tumor grades and the classification. The median overall survival for anaplastic gliomas is 48.4 months and for anaplastic astrocytoma it is 21.5 months [5], and the median survival time for patients with oligodendrogliomas is 54.8 months [6]. The most malignant of glial tumors are glioblastoma (GBM), the 1-year survival of which is 35.7% and the 5-year survival is 4.7% [7].

Country-specific data show varying results. The median overall survival time of low-grade glioma, anaplastic glioma, and glioblastoma were 78.1, 37.6, and 14.4 months in China [8], while in England and Wales, the median survival for astrocytoma, oligodendroglioma, and oligoastrocytoma was 4.6 months, 42 months, and 35.8 months, respectively [9]. The 1-, 3-, 5-, and 10-year survival rates for malignant gliomas were estimated at 63.4%, 46.2%, 39.4%, and 34.8% in South Korea [10], while the 2-year survival for Swedish people that were diagnosed with high-grade glioma was 19,5% [11]. According to the previous study, the 1-year survival for patients with malignant CNS tumors in Kazakhstan was 56.5% overall, 79.5% for lower grades, and 33.1% for grades 3 and 4 [12].

Information on the characteristics and survival rates of glial brain tumor patients from Kazakhstan is limited. Therefore, this study aimed to describe the epidemiology and characteristics of glial tumor patients in Kazakhstan based on the data from the national tertiary hospital.

## 2. Materials and Methods

### 2.1. Study Design and Population

This study retrieved 563 medical records of patients who underwent surgical treatment between January 2016 and December 2020 at the National Center for Neurosurgery. Those who were histologically diagnosed with diffuse astrocytic or oligodendroglial tumor type and had records on post-surgical treatment administration status were included in the study. After removing records that had missing data, the study was left with 335 patients. The 2007 [13] and 2016 [14] WHO Classifications of Tumors of the Central Nervous System were used to determine the type and grade of tumors, where appropriate. The diagnosis of last histological test was used as the final diagnosis. Five-year follow-up information on age at diagnosis, comorbidities, brain tumor resection surgeries, treatment received, place of residence, and nationality was collected from the national medical information system called Damumed. To determine the vital status of patients, the study gathered information from the state death registry, and patients who were reported as deceased were considered to have died from a brain tumor. Patients who were alive after the end of the follow-up period were censored.

### 2.2. Covariates

The patients’ age was stratified by quartiles into younger adults (18–30), adults (31–40), middle-aged adults (41–54), and older adults (55+). The city of patients was divided into two categories (urban and rural) by population size and development level based on the country’s administrative divisions. Regions were divided into central, north, south, east, and west based on the geographical location of the cities. The age of diagnosis corresponds to the date of the earliest magnetic resonance imaging (MRI) with a confirmed brain tumor. In cases where the MRI date was not available, the date of histological diagnosis was used instead. The time of an event was defined as a time in months from diagnosis date to the date of death or date of censoring.

After surgical removal of the tumor, patients received the following types of treatment: post-operative radiotherapy, post-operative chemotherapy, post-operative radiotherapy with chemotherapy (concurrent therapy), post-operative radiotherapy and adjuvant chemotherapy (without concurrent therapy), and post-operative concurrent and adjuvant therapy. In some cases, patients received only surgical treatment due to the low grade of the tumor, sudden death due to surgical complications, intolerance to radio or chemotherapy, or shortage of medications in a region.

### 2.3. Statistical Analysis

The 5-year survival of the patients was calculated with life tables. Descriptive data analysis included reporting of frequencies and percentages where applicable. Kaplan–Meier survival curve was built for the full cohort and Cox regression was performed with three restricted cohorts, which are low-grade tumors (Grade 1 and 2), Grade 3 tumors, and Grade 4 tumors. A multivariable Cox proportional hazard model was built for the Grade 4 tumor cohort. The proportional hazards assumption was tested with the Schoenfeld test, and treatment that was received by the patients violated the assumption (*p*-value < 0.05). The model was stratified by treatment received by the patients to account for the variable and to satisfy the assumptions of the Cox model. Statistical analysis was performed on STATA statistical software (Version 16.0; Stata Corporation, College Station, TX, USA).

## 3. Results

Among the medical records that were reviewed, 335 patients had complete and comprehensive medical information available, which was used for analysis in the study. The majority (43.3%) of the patients had Grade 4 tumors, followed by Grade 3 (27.8%), 2 (21.5%), and 1 (7.5%). The distribution of histological diagnosis within grades is demonstrated in Table 1. Grade 4 constituted glioblastoma only, while anaplastic astrocytoma was most prevalent in Grade 3. For Grade 2, diffuse astrocytoma had the highest proportion, and in Grade 1 it was pilocytic astrocytoma. The survival curves according to the grade of the tumor can be seen in Figure 1. A significant difference in survival can be seen only with Grade 4. The overall survival after 5 years of follow-up was 45.93%. Survival outcomes were the worst in the Grade 4 patients (15.64%) and best in the Grade 2 patients (73.92%). Grade 1 had a 5-year survival 68.57% and Grade 3 survival had 63.97% (Table 2).

Table 2 and Table 3 show the characteristics of patients and their survival during the 5 years stratified by these characteristics. The patients in the study had a median age of 41, mean age of 42.68 ± 13.49 years, and they were divided into four groups. The worst survival outcomes were observed among the oldest age group, where 21.83% of the sample survived 5 years since diagnosis. Most of the sample were Kazakh (69.07%) and male (56.42%). Approximately 8% of the sample had a change of diagnosis after the first histological test.

Most of the sample lived in urban areas (66.57%), and they had better 5-year survival outcomes than rural residents (49.27% vs. 38.16%). Regionally, most of the patients were from the south of the country. The best 5-year survival was among the patients from the north of the country (56.61%) and the worst was among the patients from the central region (33.79%). Approximately 27% of the sample had other comorbidities such as obesity, diabetes, other types of tumors, arterial hypertension, cardiovascular diseases, and infectious diseases. Those with comorbidities had slightly worse 5-year survival outcomes than those without the comorbidities (41.28% vs. 47.77%).

Out of 335 patients, 145 had glioblastoma. Table 3 shows that younger patients tended to have better survival rates than older patients and that females tended to have better survival rates than males. Patients who underwent radio chemotherapy and adjuvant chemotherapy had the highest 1-year OS rate (90.9%), while patients who only underwent surgery had the lowest 1-year OS rate (24.4%).

About a quarter of the patients only received the surgical intervention and did not receive further treatment. The second most common type of treatment was radiotherapy after surgery. About 18% of the sample received adjuvant chemotherapy alongside the radiation and this subset of patients had the best survival outcomes (52.2%). Almost 16% of the sample received surgery, radiotherapy, concomitant chemotherapy, and adjuvant chemotherapy. A total of 10% received concomitant chemotherapy and 6% received only chemotherapy after the surgery. The latter had the worst outcomes of the whole sample with a 35% survival rate after 5 years. The distribution of treatment methods among the grades can be seen in Table 4.

Among the low-grade patients (Grades 1 and 2), older age increased the hazard ratio of death, while other variables had no significant association. In Grade 3 patients, age was not a factor correlated with survival. However, patients of Russian nationality had almost three times worse hazards ratio when compared to Kazakhs. Grade 3 patients from urban areas on the other hand had a 73% decreased hazard ratio when compared to rural patients. In Grade 4 patients, treatment received by the patients had a significant correlation with the survival of the patients. When compared to surgery alone, all other treatment modalities decreased the risk of death with the best results among patients who underwent surgery and radiotherapy followed by adjuvant chemotherapy. For the Grade 4 patients, the multivariate Cox model stratified for the treatment method showed an increase in the risk of death with older age and a decrease in risk among female patients (Table 5).

## 4. Discussion

The prevalence of glial tumors in Kazakhstan remains unclear. Current analysis showed a higher proportion of male patients, who experience worse survival outcomes. In the US, gliomas account for 75% of all malignant brain tumors in the country with a higher prevalence in males [15]. The median age of patients with neuroepithelial tumors was 57, and 64 for glioblastomas, the latter being the most common tumor type in the subgroup [15]. Meanwhile, in our study, the average age of diagnosis for participants was 43 and the median was 41, which is considerably lower than that of the USA. However, our results are similar to those that were found in the UAE, where the average age of patients with diffuse astrocytic and oligodendroglial tumors was 38.5 years old [16]. The similarity could be explained by the demographic characteristics of the countries. Unlike the USA, where the proportion of the population over the age of 65 composes 17% of the total population [17], in Kazakhstan, the older population represents 8% of the demographic [18] and in the UAE they represent only 2% [19].

Interestingly, the survival rates of patients that were diagnosed with glial tumors in this study varied significantly from those reported by other studies. For example, in the USA, the expected survival rate for patients with glioblastoma is 5.5% [20] and 2.7% in Europe [21], while in our study 15.6% of the glioblastoma patients have survived the 5-year mark. Grade 3 tumor patients had a 64% survival rate, while in the USA, the reported survival for anaplastic oligodendroglioma and anaplastic astrocytoma were 57% and 30%, respectively [20]. Another unusual result of the study is Grade 2 tumor patients having best survival outcomes with a 5-year rate of 73.9%, while Grade 1 patients had a survival rate of 68.6%.

Such drastic differences could be attributed to the lack of molecular and genetic diagnostics in Kazakhstan. Currently, the diagnostics of brain tumors are based on histological tests and analysis, therefore, there is a possibility of a misdiagnosis that results in false survival rates. As was demonstrated by the CONCORD-3 study, the quality of tumor diagnostics and cancer registries varies greatly between countries, making it more difficult to compare the situation with CNS tumors worldwide [22]. Another factor that might influence the survival rate is the younger age of the first diagnosis of the studied sample. Older age of diagnosis is a known risk factor for worse survival outcomes [23,24], and is one of the variables that increases the risk of death in multivariable Cox regression for Grade 4 patients in the current sample.

Another factor that showed an association with an increased risk of death in high-grade tumor patients was the nationality of the patients. Kazakhstan is a multiethnic country with approximately 70% Kazakhs, 15.5% Russians, 3.2% Uzbek, and other ethnicities as reported in 2021 [25]. Compared to Kazakhs, patients of other nationalities, such as Russians, had higher crude hazard ratios of death. Differences in the incidence of CNS tumors were also noticed between races in the USA [15,26], where non-Hispanic whites had a higher incidence of gliomas and lower survival compared to other races and nationalities. There is no clear explanation for differences in incidence and survival. Some of the influential factors might be socio-economic status, geography, access to healthcare, or genetics.

There was also a difference in the length of survival between patients from rural and urban cities; those who lived in bigger cities had a 49.3% overall survival rate, while those from rural regions had a survival rate of 38.2%. Moreover, in Grade 3 tumor patients, those from urban cities had a 73% lower hazard ratio of death when compared to the rural patients. Similar findings were reported in Finland, where glioblastoma patients that were treated in high-volume hospitals in the capital city had better survival rates [27], as well as in Sweden which reported better outcomes for patients from the Stockholm area [11]. In Kazakhstan, healthcare is available for every citizen for free through Compulsory Social Health Insurance [28]. Even though all the patients that were included in the study had undergone resection of the tumor in the capital city at the National Center for Neurosurgery, their anticancer therapy was conducted and administered at their place of residence. There is a possibility of delay in access to treatment as well as lower quality of treatment received by the rural patients. Another important factor that has been demonstrated to influence survival in glioblastoma patients is their socio-economic status [29], which could be directly related to the area of their residency. However, such information is unavailable in this study, and more research is required to prove such a discrepancy. As was pointed out in the study that was performed in the Netherlands, other patient factors might be a confounder resulting in such differences [30].

Another unusual pattern in our study was the survival rate of Grade 1 patients, which was lower than that of patients with Grade 2 tumors (69% vs. 74%) and considerably lower than rates in other regions such as the USA and Europe (94% [20] and 80.5% [21], respectively). The observed results possibly stem from 40% of the Grade 1 sample receiving chemotherapy (CT)or radiotherapy (RT) treatment. CT and RT are not commonly recommended courses of action for patients with Grade 1 tumors [20], and, as was observed in the Turkoglu et al. study, they may lead to worse OS rates [31]. However, it should be noted that the extent of resection in the studied patients is unknown, and limited resection could be the reason for CT/RT treatment prescription as well as lower survival rates.

Among Grade 4 patients, the best outcomes were observed in those who had undergone surgical treatment followed by radiotherapy and adjuvant chemotherapy. Although radio chemotherapy (RCT) followed by adjuvant chemotherapy is the recommended Stupp protocol for the treatment of Grade 4 glial tumor patients [32], in this sample, RT followed by adjuvant CT showed better survival outcomes. In Grade 3 patients, the treatment they received did not affect their survival significantly, although chemoradiation is the recommended course of action [20]. With each update in the WHO Classification of Tumors of the Central Nervous System, advances in molecular diagnostics has become an integral part of the proper diagnosis and treatment of tumors [33]. The lack of such advanced diagnostic tools in Kazakhstan might explain the discrepancy between the obtained results with results from other studies. This once again points to the urgency of the introduction of the genetic component in the diagnosis of glial tumors in Kazakhstan.

This study has several limitations. First, although the data were collected from a national hospital that accepts patients from all over the country, the findings are not generalizable to the whole population. The sample size needs to be expanded with a random selection from hospitals and registries across the country, and a better representation of lower-grade tumor patients is required. Moreover, due to the retrospective nature of the data, several important factors were not adjusted for in the analysis, such as the extent of resection, socio-economic status, Karnofsky score, length and dosage of radio and chemotherapy, and others. Finally, the lack of molecular and genetic diagnostics in the country makes the diagnosis of the patients imprecise and does not allow for complete adherence to the WHO criteria for tumor classification.

## 5. Conclusions

The five-year survival of glial tumor patients in Kazakhstan is 45.93% and patients were relatively young at the age of diagnosis. The survival rates of glioblastoma patients were higher in Kazakhstan than in other regions of the world. However, Grade 1 patients were observed to have lower survival rates than expected. This could be attributed to the inaccurate diagnostics of the tumor subtypes, as the country lacks the necessary molecular and genetic diagnostic facilities. The study revealed a significant rise in the risk of mortality among patients belonging to ethnicities other than Kazakh and those residing in rural areas. However, to obtain a comprehensive understanding of the prevalence and associated risk factors of glial tumors in Kazakhstan, further investigations are warranted on a larger scale that can accurately represent the entire country.

## Figures and Tables

**Figure 1 biomedicines-11-00886-f001:**
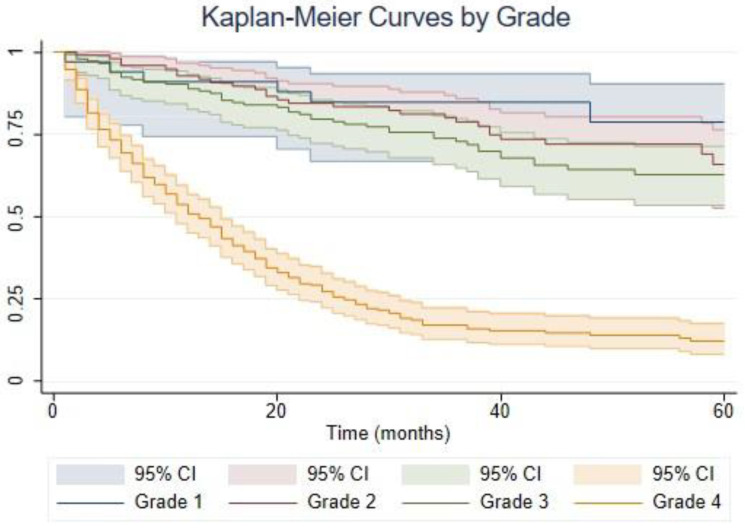
Kaplan–Meier plot by grades. CI-confidence interval.

**Table 1 biomedicines-11-00886-t001:** Distribution of histological diagnosis within glial tumor grades.

WHO Grade	Histological Diagnosis	n	% Within Grade	% Total
1	desmoplastic infantile astrocytoma and ganglioglioma	1	4	0.30
	ganglioglioma	1	4	0.30
	pilocytic astrocytoma	23	98	6.87
2	diffuse astrocytoma	53	73.61	15.82
	oligoastrocytoma	12	16.67	3.58
	oligodendroglioma	5	6.94	1.49
	pleomorphic xanthastrocytoma	2	2.78	0.60
3	anaplastic astrocytoma	38	40.86	11.34
	anaplastic ganglioglioma	1	1.08	0.30
	anaplastic oligoastrocytoma	23	24.73	6.87
	anaplastic oligodendroglioma	28	30.11	8.36
	anaplastic pleomorphic xanthastrocytoma	3	3.23	0.90
4	glioblastoma	145	100	43.28
	Total	335		100

**Table 2 biomedicines-11-00886-t002:** Demographic and clinical characteristics of patients and their survival.

Variable		N (%)	1 Year OS (95% CI)	2 Years OS (95% CI)	5 Years OS (95% CI)
Total		335 (100)	79.1 (74.35–83.08)	63.53 (58.09–68.46)	45.93 (39.67–51.95)
Age (Me—41, mean—42.68 ± 13.49)	18–30	78 (23.28)	89.74 (80.54–94.73)	83.19 (72.82–89.88)	69.28 (56.42–70.03)
	31–40	81 (24.18)	88.89 (79.74–94.06)	73.97 (62.93–82.18)	56.23 (41.49–68.61)
	41–54	98 (29.25)	77.55 (67.94–84.60)	58.43 (47.90–67.55)	36.77 (25.72–47.83)
	≥55	78 (23.28)	60.26 (48.53–70.12)	39.30 (28.45–49.96)	21.83 (12.39–32.99)
Nationality	Kazakh	230 (69.07)	80.43 (74.69–85)	66.21 (59.67–71.95)	47.45 (39.68–54.82)
	Russian	54 (16.22)	81.48 (68.32–89.58)	64.43 (50.05–75.64)	45.1 (30.22–58.87)
	Other	49 (14.71)	69.39 (54.45–80.28)	50.75 (36.05–63.7)	40.84 (26.64–54.55)
Sex	Male	189 (56.42)	70.81 (65.51–75.45)	51.8 (46.17–57.12)	37.73 (31.87–43.56)
	Female	148 (43.66)	71.73 (65.58–76.97)	57.55 (51–63.56)	39.14 (32–46.19)
Change of diagnosis	Yes	27 (8.06)	96.30 (76.49–99.47)	81.19 (60.56–91.71)	56.95 (34.12–74.47)
	No	308 (91.94)	77.60 (72.52–81.86)	61.98 (56.27–67.17)	45.55 (39.12–51.75)
City type	Rural	112 (33.43)	76.79 (67.81–83.56)	62.25 (52.54–70.52)	38.16 (26.53–49.69)
	Urban	223 (66.57)	80.27 (74.41–84.92)	64.17 (57.45–70.11)	49.27 (41.89–56.22)
Region	Central	68 (20.42)	69.12 (56.67–78.64)	49.37 (36.94–60.63)	33.79 (21.97–46)
	East	19 (5.71)	78.95 (53.19–91.53)	73.68 (47.89–88.1)	46.68 (21.68–68.41)
	North	85 (25.53)	78.82 (68.52–86.09)	68.07 (56.99–76.87)	56.61 (44.91–66.73)
	South	95 (28.53)	80 (70.46–86.74)	64 (53.45–72.77)	44.65 (31.77–56.71)
	West	66 (19.82)	87.88 (77.22–93.75)	67.29 (54.35–77.31)	47.28 (32.64–60.58)
Treatment method	Only surgery	86 (25.67)	54.65 (43.56–64.45)	46.33 (35.53–56.45)	46.33 (35.53–56.45)
	Chemotherapy	20 (5.97)	75.00 (49.99–88.75)	35.00 (15.66–55.19)	35.00 (15.66–55.19)
	Radiotherapy	84 (25.07)	85.71 (76.22–91.62)	75.85 (65.09–83.69)	46.96 (33.03–59.71)
	Radio chemotherapy	32 (9.55)	78.13 (59.52–88.92)	56.25 (37.59–71.30)	44.74 (26.51–61.44)
	Radio chemotherapy and adjuvant chemotherapy	53 (15.82)	92.45 (81.13–97.10)	70.81 (56.31–81.27)	43.74 (27.17–59.17)
	Radiotherapy and adjuvant chemotherapy	60 (17.91)	95.00 (85.29–98.36)	78.19 (65.41–86.70)	52.20 (38.18–64.97)
Comorbidity	Yes	89 (26.57)	69.66 (58.97–78.08)	54.69 (43.74–64.39)	41.28 (30.18–52.03)
	No	246 (73.43)	82.52 (77.17–86.73)	66.72 (60.41–72.26)	47.77 (40.31–54.83)
Grade	1	25 (7.46)	88.00 (67.26–95.96)	80.00 (58.44–91.15)	68.57 (37.51–86.49)
	2	72 (21.49)	93.06 (84.12–97.05)	86.06 (75.64–92.24)	73.92 (61.12–83.07)
	3	93 (27.76)	92.47 (84.86–96.34)	86.94 (78.14–92.36)	63.97 (50.72–74.52)
	4	145 (43.28)	62.07 (53.65–69.40)	34.24 (26.56–42.06)	15.64 (9.34–23.41)

OS—overall survival, me—median, CI—confidence interval.

**Table 3 biomedicines-11-00886-t003:** Demographic and clinical characteristics of patients with glioblastoma and their survival.

Variable		N (%)	1 Year OS (95% CI)	2 Years OS (95% CI)	5 Years OS (95% CI)
Total		145 (100)	57.9 (50.4–66.6)	33.4 (26.5–42.1)	15.9 (10.1–25.0)
Age	18–30	19 (13.10)	73.7 (56.3–96.4)	52.1 (33.7–80.5)	24.4 (10.0–59.7)
	31–40	21 (14.48)	81.0 (65.8–99.6)	42.9 (26.2–70.2)	30.0 (14.7–61.0)
	41–54	49 (33.79)	53.1 (40.8–69.0)	34.7 (23.6–50.9)	13.3 (4.6–38.4)
	≥55	56 (38.62)	48.2 (36.8–63.2)	23.0 (14.2–37.2)	6.4 (1.9–22.3)
Nationality	Kazakh	96 (66.21)	63.5 (54.6–73.9)	37.1 (28.6–48.3)	16.6 (9.5–29.0)
	Russian	25 (17.24)	56.0 (39.6–79.3)	32.0 (18.1–56.7)	16.0 (6.0–42.8)
	Other	24 (16.55)	37.5 (22.4–62.9)	20.0 (8.8–45.4)	15.0 (5.5–40.6)
Sex	Male	86 (59.31)	52.3 (42.8–64.0)	27.9 (19.9–39.2)	16.3 (9.4–28.4)
	Female	59 (40.69)	66.1 (55.1–79.4)	40.9 (29.8–56.1)	12.8 (5.2–31.5)
Change of diagnosis	Yes	16 (11.03)	87.5 (72.7–100.0)	61.9 (41.9–91.4)	45.1 (25.0–81.4)
	No	129 (88.97)	54.3 (46.3–63.6)	29.8 (22.9–39.0)	12.4 (7.0–21.9)
City type	Rural	46 (31.72)	56.5 (43.9–72.8)	33.8 (22.3–51.0)	8.4 (1.7–41.2)
	Urban	99 (68.28)	58.6 (49.6–69.1)	33.1 (25.0–43.9)	18.1 (11.4–28.8)
Region	Central	36 (24.83)	50.0 (36.1–69.3)	24.7 (13.9–43.9)	12.3 (5.0–30.4)
	East	4 (2.76)	25.0 (4.6–1.0)	-	-
	North	39 (26.90)	59.0 (45.4–76.6)	35.5 (23.2–54.4)	25.4 (13.5–47.8)
	South	38 (26.21)	65.8 (52.3–82.7)	38.8 (25.9–58.2)	15.6 (6.3–38.4)
	West	28 (19.31)	60.7 (45.1–81.8)	35.7 (21.7–58.7)	12.3 (3.9–39.1)
Treatment method	Only surgery	41 (28.28)	24.4 (14.2–41.8)	17.1 (8.7–33.5)	17.1 (8.7–33.5)
	Chemotherapy	13 (8.97)	53.9 (32.6–89.6)	7.7 (1.2–50.6)	-
	Radiotherapy	21 (14.48)	47.6 (30.4–74.6)	37.5 (21.4–65.6)	11.3 (2.2–59.0)
	Radio chemotherapy	18 (12.41)	61.1 (42.3–88.3)	38.9 (21.8–69.4)	33.3 (17.3–64.1)
	Radio chemotherapy and adjuvant chemotherapy	30 (20.69)	86.7 (75.3–99.7)	41.5 (26.7–64.5)	5.8 (0.9–36.6)
	Radiotherapy and adjuvant chemotherapy	22 (15.17)	90.9 (79.7–100.0)	59.1 (41.7–83.7)	25.8 (11.9–56.1)
Comorbidity	Yes	43 (29.66)	46.5 (33.8–64.1)	25.1 (14.9–42.4)	14.4 (6.3–32.8)
	No	102 (70.34)	62.7 (54.0–72.9)	36.9 (28.6–47.7)	16.4 (9.4–28.5)

OS—overall survival, me—median, CI—confidence interval.

**Table 4 biomedicines-11-00886-t004:** Distribution of treatment methods by grades.

Treatment	Grade 1, n (%)	Grade 2, n (%)	Grade 3, n (%)	Grade 4, n (%)	Total, n (%)
Only surgery	15 (60)	16 (22.22)	14 (15.05)	41 (28.28)	86 (25.67)
Chemotherapy	1 (4)	-	6 (6.45)	13 (8.97)	20 (5.97)
Radiotherapy	7 (28)	24 (33.33)	32 (34.41)	21 (14.48)	84 (25.07)
Radio chemotherapy	-	7 (9.72)	7 (7.53)	18 (12.41)	32 (9.55)
Radio chemotherapy and adjuvant chemotherapy	-	7 (9.72)	16 (17.2)	30 (20.69)	53 (15.82)
Radiotherapy and adjuvant chemotherapy	2 (8)	18 (25)	18 (19.36)	22 (15.17)	60 (17.91)
Total	25 (100)	72 (100)	93 (100)	145 (100)	335 (100)

**Table 5 biomedicines-11-00886-t005:** Crude and adjusted hazards ratio.

	Low Grade		Grade 3		Grade 4			
Variable	CHR (95% CI)	*p*-Value	CHR (95% CI)	*p*-Value	CHR (95% CI)	*p*-Value	AHR (95% CI) *	*p*-Value
Age	1.06 (1.03–1.1)	<0.001	1.02 (0.99–1.05)	0.24	1.03 (1.01–1.04)	<0.001	1.02 (1.01–1.04)	0.005
Sex								
Male	Ref		Ref		Ref		Ref	
Female	0.89 (0.39–2.04)	0.79	0.57 (0.25–1.28)	0.17	0.81 (0.56–1.18)	0.28	0.52 (0.33–0.81)	0.004
Nationality								
Kazakh	Ref		Ref		Ref		-	-
Russian	-	-	2.89 (1.17–7.14)	0.02	1.22 (0.75–1.98)	0.43	-	-
Other	1.18 (0.44–3.17)	0.75	2.21 (0.64–7.66)	0.21	1.57 (0.95–2.57)	0.08		
Comorbidity								
No	Ref		Ref		Ref		-	-
Yes	0.96 (0.36–2.58)	0.93	1.61 (0.72–3.62)	0.25	1.35 (0.91–2.01)	0.14	-	-
City type								
Rural	Ref		Ref		Ref		-	-
Urban	1.35 (0.56–3.29)	0.67	0.27 (0.12–0.59)	0.001	0.97 (0.65–1.43)	0.86	-	-
Treatment method								
Only surgery	Ref		Ref		Ref		-	-
Chemotherapy	-	-	0.43 (0.05–3.66)	0.44	0.78 (0.4–1.53)	0.47	-	-
Radiotherapy	1.14 (0.41–3.15)	0.8	0.71 (0.25–2.06)	0.53	0.53 (0.29–0.96)	0.035	-	-
Radio chemotherapy	1.13 (0.23–5.44)	0.88	1.5 (0.35–6.32)	0.58	0.39 (0.2–0.75)	0.005	-	-
Radio chemotherapy and adjuvant chemotherapy	0.6 (0.07–4.91)	0.64	-	-	0.41 (0.24–0.7)	0.001	-	-
Radiotherapy and adjuvant chemotherapy	1.01 (0.32–3.17)	0.99	0.7 (0.21–2.3)	0.56	0.33 (0.18–0.6)	<0.001	-	-

*-stratified for treatment method, low grade—Grade 1 and 2, CHR—crude hazard ratio, AHR — adjusted hazard ratio, CI—confidence interval.

## Data Availability

All the data that were collected and used for this study are available for public viewing via the link https://doi.org/10.5281/zenodo.7098789 (accessed on 6 March 2023).

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
