# Peer review of "The Current State of Adult Glial Tumor Patients’ Care in Kazakhstan: Challenges in Diagnosis and Patterns in Survival Outcomes"

_biomedicines, 2023, doi:10.3390/biomedicines11030886_

Round 1
Reviewer 1 Report
The authors chose to discuss a relevant topic as it related to the outcomes for those with glial tumors in Kazakhstan. However, the terminology being provided is confounding, as grade glioblastoma is considered a disease unto itself as it is a grade 4 astrocytoma.
The Kaplan Meier curves do not provide any novel insights into the mortality status of grade 4 disease.
This reviewers questions accuracy of the sampled population, and whether these tumors were properly categorized into astrocytomas as opposed to all gial tumors.
Author Response
Dear Reviewer,
We would like to express our sincere gratitude to you for your insightful comments, constructive feedback, and suggestions that significantly improved the quality of our research. Please find a point-by-point response to the comments below.
Comment: The authors chose to discuss a relevant topic as it related to the outcomes for those with glial tumors in Kazakhstan. However, the terminology being provided is confounding, as grade glioblastoma is considered a disease unto itself as it is a grade 4 astrocytoma.
Response: We appreciate your comments regarding the possible misclassification of glioblastoma and grade 4 astrocytoma cases in our study. However, all astrocytomas in our data were grade 3, and grade 4 tumors includes only glioblastoma. This suggests that the misclassification of glioblastoma as grade 4 astrocytoma or vice versa is less likely to have occurred in our study.
Comment: The Kaplan Meier curves do not provide any novel insights into the mortality status of grade 4 disease.
Response: While Kaplan-Meier curves do not offer new insights into the mortality rates of grade 4 diseases, they do reveal unusual patterns in those of grades 1 and 2. This could suggest that misdiagnosis may be occurring due to the absence of genetic testing.
Comment: This reviewers questions accuracy of the sampled population, and whether these tumors were properly categorized into astrocytomas as opposed to all gial tumors.
Response: Our study categorized brain tumors based on the final diagnosis recorded in their medical records, which led to the use of the term "glioblastoma" for almost half of the patients in the study. It should be noted that this approach may have limitations due to the potential for misdiagnosis, which was acknowledged and addressed in the "limitations" section of the study.
Glioblastoma was classifies as astrocytic tumor in WHO 2007 classification and refer to diffuse astrocytic and oligodendroglial tumours in the 2016 WHO classification. Our study used “glioblastoma” terminology as patients were diagnosed based on the histological analysis. Nevertheless, we recognize that histological analysis alone may not always be sufficient to accurately differentiate the tumor types, and molecular analysis is becoming increasingly important in this regard.
Comment: Extensive editing of English language and style required
Thank you again for your thoughtful feedback, and we welcome any further comments or questions you may have.
Response: English language editing has been done.
Reviewer 2 Report
The paper "The current state of adult glial tumor' care patients in Kazakhstan: challenges in diagnosis and patterns in survival outcomes" by Aisha Babi et al. deals with glial tumors and face the critical issue of financial toxicity. This type of tumors has become the leading cause of cancer death among males aged birth to 39 years and females aged 19 years. Malignant glioma is a devastating type of brain and nervous system tumor because of its high malignancy, very high mortality rate and risk of recurrence, and the enormous burden it places on society and families. It is responsible for the majority of deaths in patients with primary brain tumors. The incidence and mortality of glioma increase significantly with advancing age. With the aging trend in society, the burden associated with glioma could become a huge challenge. Although no global population-based epidemiological survey of glioma has been reported, the incidence of glioma varies geographically. The incidence of glioma in American and Northern European populations has been found to be higher than in Asian populations, doubling. Different incidence trends have also been found regionally. A Japanese study reported that the incidence rate of malignant glioma increased significantly among the elderly from 1989 to 2008, while CBTRUS reported that the incidence of glioma in people over the age of 40 in the United States remained relatively stable from 2000 to 2016. In addition, glioma incidence varies significantly by age, sex, race, and other factors. Compared with the relatively low incidence of glioma, the annual number of deaths caused by glioma deserves attention. The death rate is a compelling indicator to reflect progress against glioma. However, few studies have focused on glioma mortality.
For many decades, the extent of tumor resection and overall survival were almost the only parameters by which treatment success and quality of care of glioma patients were measured. With the advent of new systemic therapeutic approaches and new imaging technologies, and the increasing understanding of brain connectivity and molecular mechanisms of glioma disease, patient-centered individualized therapy of glioma has evolved. Today, therefore, many patients, especially those with IDH-mutated gliomas with a more favorable prognosis, are faced with chronic disease rather than an end-of-life perspective soon after tumor diagnosis. As a result, the quality of care for glioma patients has been brought into focus, mainly encompassing high-quality shared decision-making, excellent performance, outcome measures, and accessibility to treatment. However, in terms of high-level techniques, complex quality and process management, and a holistic approach to patients, considering their biopsychosocial situation, the quality of care in this context is still relegated to high-income countries. This is even more true for brain tumor patients, for whom the highest incidence rates were found in countries with high levels of sociodemographic index, reflecting the lack of accessibility to advanced and expensive imaging technologies and advanced neurological and neurosurgical services, as the authors of this study correctly noted. This aspect lies within financial toxicity and is something that needs to be urgently addressed to avoid increasing socioeconomic disparity in access to care. The complaint made by the authors about the observed differences based on geographic and urban versus rural population distribution that could be attributed to the lack of molecular and genetic diagnostics in Kazakhstan is reliable given what is happening globally and is very serious. Basing the diagnosis of brain tumors only on histological examinations and analysis, with the possibility of misdiagnosis resulting in false survival rates requires a thorough analysis. The authors also pointed out another unusual finding regarding the survival rate of patients with grade 1 tumors lower than that of patients with grade 2 tumors and considerably lower than rates in other regions such as the United States and Europe.
The observed results could be due to the fact that 40% of grade 1 tumors received treatment with CT or RT which is absolutely not the clinical indication for treatment for this pathology. The authors report as a possible justification that the CH RT indication may be due to imperfect knowledge of the extent of resection as limited resection in the patients studied would lead to the prescription of CT/RT treatment to increase local tumor control but leading to serious risks of undue exposure.
Authors should solicit adherence with their data reported in the study to international databases to share results and adhere to established protocols.
Authors are invited to explore this further in the discussion section.
The increasing digitization and integration of medical records over the past 50 years has led to the accumulation of huge collections of clinical data on patients with central nervous system tumors, including low- and high-grade gliomas. Some of these data have been made widely available for research in the form of open-access or subscription-based clinical databases. More recently, with the advancement of genomic sequencing technology, the acquisition of cancer genomic data has become more economically feasible and widely accessible, enabling the development of robust online genomics in many areas of the world. In addition, due to population growth, aging, and the environmental and socioeconomic situation of health care-including rising inflation and shortages of qualified health care workers-maintaining the quality of care for glioma patients will be a challenge in the future. databases to follow suit. At the forefront of this data revolution are a growing number of multi-institutional consortia and nationally funded data repositories striving to pool data and resources to address key clinical questions and advance the care of cancer patients. The unprecedented amount of data now available has created an opportunity to study human diseases, such as glioma, from a new perspective by combining newly available genomic data with an ever-growing compendium of clinical knowledge. The literature lacks a comprehensive review of the clinical and molecular data available for the study of glioma.
The wide variety of databases and registries available for clinical and genomic research on glioma is a powerful tool to manage this critical topic. With databases and registries covering multiple countries, cohorts, years, clinical variables, and molecular data types, these resources provide data that can answer a wide range of research questions. Strengths include large sample sizes, the inclusion of several clinical variables, and the availability of molecular data that are expensive to collect covering many disease states. Limitations include heterogeneity of data and tissues, completeness of variables, barriers to access, and lack of glioma-specific variables in many databases.
Previously, clinical and genomic studies have been undertaken based on locally collected data on a limited sample size due to logistical and financial limitations. A relatively small cohort of researchers with access to patient data and the means to analyze them were able to undertake clinically meaningful studies. With the advent of a growing number of national databases and consortia, vast collections of clinical information are now available to anyone with a scientific question and an Internet connection. In the study of glioma, these databases provide large multinational data sets that can be used to study outcomes, the molecular composition of glioma, and even the clinical and molecular correlates of different treatments. These clinical databases could provide the data needed to help feed the externally controlled studies that have been proposed as a means of advancing glioma research. Large multinational databases offer opportunities for growth. Although small glioma-specific databases, such as GLASS and IVY GAP, provide specific, high-quality molecular information, small sample sizes and lack of long-term follow-up are currently limitations. Expanding large datasets such as SEER, NCDB, and CBTRUS to include glioma-specific variables would produce robust annual cohorts of detailed glioma data, which could improve future research opportunities on large databases in glioma.
Furthermore, although this study has outlined the sources and content of available public data on glioma, researchers are often unaware of what public data are available or, more often, how to access them. In the current paradigm, databases provide a general overview of their contents, but the specific contents are often not fully understood until the researcher obtains permission to access the data. Databases and consortia should bridge this information gap by clearly publishing their contents with explicit enumeration of variables included, sample sizes, and disease states covered. New resources should be highlighted in peer-reviewed journals and professional conferences. Data access pipelines must be optimized to facilitate data access by researchers while maintaining data security standards and patient privacy.
Author Response
Dear Reviewer,
We are grateful for your time, effort, and expertise, which have contributed immensely to our research. We would like to express our sincere gratitude to you for your valuable suggestions regarding the databases. We appreciate your insights and suggestions, which we will definitely consider in future studies to enhance the research's scope and relevance.
Comment: Authors are invited to explore this further in the discussion section.
Response: the possible explanation of unusual patterns in survival rates in grades 1 and 2 is given in a discussion section, lines 245-253.
Reviewer 3 Report
Authors should prepare a table describing the survival of patients with glioblastoma by age, nationality, sex, treatment method, city type, nationality. Same as table 2 describing patients with all brain tumors.
The authors should add a tables describing how often (cases per 100,000 population) brain tumors (and glioblastoma) of various grades occur in terms of age, gender, place of residence, etc.
Authors should show the data in Table 2 (N %) converted per 100,000 population.
Author Response
Comment: Authors should prepare a table describing the survival of patients with glioblastoma by age, nationality, sex, treatment method, city type, nationality. Same as table 2 describing patients with all brain tumors.
Response: a Table describing the survival of patients with glioblastoma by age, nationality, sex, treatment method, city type, and nationality was prepared and given in lines 143-144. The description was added within the text, lines 145-149.
Comment: The authors should add a tables describing how often (cases per 100,000 population) brain tumors (and glioblastoma) of various grades occur in terms of age, gender, place of residence, etc.
Response:
We appreciate your valuable feedback on our article. We agree with your suggestion to provide a table describing the prevalence of brain tumors and glioblastoma, but we would like to clarify that the data we used in our study were collected from a single hospital over a period of 5 years, and we excluded patients with insufficient data. Therefore, estimating the prevalence rate of brain tumors and glioblastoma for the population based on our study data may not be appropriate.
Comment: Authors should show the data in Table 2 (N %) converted per 100,000 population.
Response: We would like to highlight that our study is a retrospective study and the data were collected from a single hospital. The data we have used in our study may not be representative of the entire population, and thus converting data per 100,000 population would not be appropriate. Moreover, we excluded patients with incomplete data to ensure the quality and accuracy of our findings. This further limits the generalizability of our study results.
Round 2
Reviewer 1 Report
The reviewer thanks the authors for their kind comments. Unfortunately by their own volition, the study suffers from misdiagnosis , which limits the study design.
:
Our study categorized brain tumors based on the final diagnosis recorded in their medical records, which led to the use of the term "glioblastoma" for almost half of the patients in the study. It should be noted that this approach may have limitations due to the potential for misdiagnosis, which was acknowledged and addressed in the "limitations" section of the study
Reviewer 3 Report
The authors revised the article in accordance with the reviewer's guidelines.